# Clusters of polymorphic transmembrane genes control resistance to schistosomes in snail vectors

Jacob A Tennessen[1,2]*, Stephanie R Bollmann[2], Ekaterina Peremyslova[2], Brent A Kronmiller[2,3], Clint Sergi[2], Bulut Hamali[2], Michael Scott Blouin[2]

[1]Department of Immunology and Infectious Diseases, Harvard T. H. Chan School of Public Health, Boston, United States; [2]Department of Integrative Biology, Oregon State University, Corvallis, United States; [3]Center for Genome Research and Biocomputing, Oregon State University, Corvallis, United States

**Abstract** Schistosomiasis is a debilitating parasitic disease infecting hundreds of millions of people. Schistosomes use aquatic snails as intermediate hosts. A promising avenue for disease control involves leveraging innate host mechanisms to reduce snail vectorial capacity. In a genome-wide association study of *Biomphalaria glabrata* snails, we identify genomic region PTC2 which exhibits the largest known correlation with susceptibility to parasite infection (>15 fold effect). Using new genome assemblies with substantially higher contiguity than the *Biomphalaria* reference genome, we show that PTC2 haplotypes are exceptionally divergent in structure and sequence. This variation includes multi-kilobase indels containing entire genes, and orthologs for which most amino acid residues are polymorphic. RNA-Seq annotation reveals that most of these genes encode single-pass transmembrane proteins, as seen in another resistance region in the same species. Such groups of hyperdiverse snail proteins may mediate host-parasite interaction at the cell surface, offering promising targets for blocking the transmission of schistosomiasis.

**\*For correspondence:**
jtennessen@hsph.harvard.edu

**Competing interests:** The authors declare that no competing interests exist.

## Introduction

Schistosomiasis is a chronic and debilitating disease suffered by over 200 million people worldwide (*Evan Secor, 2014*; *Lo et al., 2018*). It is caused by infection with schistosome trematode parasites that are transmitted by aquatic snails (*Lo et al., 2018*). Infection can be treated with regular doses of a single drug, praziquantel (*Doenhoff et al., 2009*). However, it has become increasingly clear that mass drug administration alone will not adequately control schistosomiasis, and that successful elimination of transmission requires intervention at the snail stage (*Lo et al., 2018*; *Sokolow et al., 2018*).

Immunogenetic interactions between snails and schistosomes represent a crucial stage in the parasite life cycle that can be targeted to block transmission. Parasite resistance is highly heritable in snails, and there is substantial strain-by-strain interaction between hosts and parasites (*Richards and Shade, 1987*; *Richards et al., 1992*; *Knight et al., 1999*; *Webster et al., 2004*; *Webster and Woolhouse, 1998*; *Theron et al., 2014*). Genetically diverse parasite cultures can infect nearly all snails, while bottlenecked laboratory strains of parasite can only infect a subset (*Theron et al., 2008*), suggesting a 'trench warfare' model in which numerous alleles are maintained in both host and parasite populations because each matches a different phenotype in the other species (*Seger, 1988*; *Stahl et al., 1999*). There are likely to be snail genes with large effects on resistance to particular parasite genotypes (*Lewis et al., 2001*). Finding these genes will open two potential avenues for disease mitigation. First, they may uncover mechanisms of infection by the parasite that could be therapeutically targeted. Second, they may facilitate genetic manipulation of wild snail populations so as

**eLife digest** Schistosomiasis is a widespread parasitic disease, affecting over 200 million people in tropical countries. It is caused by schistosome worms, which are carried by freshwater snails. These snails release worm larvae into the water, where they can infect humans – for example, after bathing or swimming.

Treatment options for schistosomiasis are limited. Eliminating the freshwater snails is one way to control the disease, but this is not always effective in the long term and the chemicals used can also harm other animals in the water.

Another way to manage schistosomiasis could be to stop the worms from infecting their snail host by breaking the parasites' life cycle without killing the snails. It is already known that some snails are naturally resistant to infection by some strains of schistosomes. Since this immunity is also inherited by the offspring of resistant snails, there is likely a genetic mechanism behind it. However, very little else is known about any genes that might be involved. Tennessen et al. therefore set out to identify what genes were responsible for schistosome resistance and how they worked.

The experiments used a large laboratory colony of snails, whose susceptibility to schistosome infection varied among individual animals. To determine the genes behind this variation, Tennessen et al. first searched for areas of DNA that also differed between the immune and infected snails. Comparing genetic sequences across over 1,000 snails revealed a distinct region of DNA that had a large effect on how likely they were to be infected.

This section of DNA turned out to be highly diverse, with different snails carrying varying numbers and different forms of the genes within this region. Many of these genes appear to encode proteins found on the surface of snail cells, which could affect whether snails and worms can recognize each other when they come into contact. This in turn could determine whether or not the worms can infect their hosts.

These results shed new light on how the snails that carry schistosomes may be able to resist infections. In the future, this knowledge could be key to controlling schistosomiasis, either by releasing genetically engineered, immune snails into the wild (thus making it harder for the parasites to reproduce) or by using the snails' mechanism of resistance to design better drug therapies.

to reduce parasite transmission by eliminating alleles that permit infection by certain schistosome genotypes (*Theron et al., 2014*; *Reardon, 2016*; *Famakinde, 2018*; *Maier et al., 2019*).

For invertebrates, highly strain-specific responses to parasites remain unexplained (*Schmid-Hempel, 2005*; *Schulenburg et al., 2007*), as these taxa lack the combinatorial immune system found in vertebrates with its vast and adjustable repertoire of recognition molecules. Invertebrate resistance to macroparasites or parasitoids often involves large-effect loci (*Carton et al., 2005*), but most of these are poor candidates for mediating strain specificity, as they reflect generic enhanced defenses conveying a constitutive fitness cost (*Kraaijeveld and Godfray, 1997*; *Webster and Woolhouse, 1999*; *Koella and Boëte, 2002*), or encode signaling molecules (*Hita et al., 2006*) or effectors (*Goodall et al., 2006*) rather than recognition molecules. Strain specificity may be conveyed by suites of highly diverse host genes that act synergistically, especially if these interact with similarly varying and coevolving sets of parasite genes to mediate host-parasite recognition (*Schmid-Hempel, 2005*; *Schulenburg et al., 2007*; *Cerenius and Söderhäll, 2013*). Such loci will not necessarily produce a large phenotypic signal unless several are clustered in the same genomic region.

The neotropical snail *Biomphalaria glabrata* has been the focus of recent efforts to develop genomic resources for schistosomiasis vector biology, although the 916 Mb, repetitive genome remains poorly assembled (reference genome *BglaB1* N50 = 48 kb; *Adema et al., 2017*; *Tennessen et al., 2017*). *B. glabrata* snails can be readily challenged with *Schistosoma mansoni* miracidia under controlled laboratory conditions, with successful infections diagnosed by subsequent shedding of cercariae (*Bonner et al., 2012*). To date, four genomic regions have been identified in which allelic variation influences resistance to infection by *S. mansoni* (*Knight et al., 1999*; *Goodall et al., 2006*; *Tennessen et al., 2015a*; *Tennessen et al., 2015b*). An F2 mapping cross between resistant *B. glabrata* strain BS-90 and a susceptible strain yielded two RAPD markers linked to resistance (*Knight et al., 1999*). One of those was subsequently aligned to a contig on Linkage Group (LG) XII

(*Tennessen et al., 2017*), although the identities of any candidate genes to which it may be linked remain unclear (the other could not be uniquely mapped). Two other genomic regions (sod1 and RADres; *Goodall et al., 2006*; *Tennessen et al., 2015a*) influence *S. mansoni* infection in *B. glabrata* population 13–16-R1 which is admixed from Caribbean and Brazilian populations and has been maintained free from parasites for several decades as a large laboratory population with substantial segregating variation. However, together those two regions explain only 7% of the variance in resistance in 13-16-R1, suggesting that other resistance loci remain undiscovered in this population. In *B. glabrata* from Guadeloupe, the Guadeloupe Resistance Complex (GRC) shows an 8-fold effect on the odds of *S. mansoni* infection (*Tennessen et al., 2015b*; *Allan et al., 2017*) via a hemocyte-mediated mechanism (*Allan et al., 2018a*) that also affects the proteome (*Allan et al., 2019*) and microbiome (*Allan et al., 2018b*). Seven clustered GRC genes encode hyperdiverse single-pass transmembrane (TM1) proteins that appear to recognize parasite-associated molecules (*Tennessen et al., 2015b*), likely including saccharides (*Allan and Blouin, 2018*).

Here, we use a genome-wide association study to pinpoint a new resistance region in snails, with a very large (over 15-fold) effect on odds of infection by schistosomes. It comprises a cluster of highly polymorphic transmembrane genes, and as GRC (=PTC1) was the first such cluster described in *Biomphalaria*, we designate this second region as Polymorphic Transmembrane Cluster 2 (PTC2). Using PacBio, we have vastly improved the assembly of the *B. glabrata* genome, allowing us to fully characterize the chromosomal vicinity of PTC2. Transcriptomic data show that, like GRC, PTC2 harbors exceptionally divergent suites of TM1 genes, suggestive of coevolutionary dynamics. These results support a general immunogenetic scenario in which clusters of highly polymorphic TM1 genes mediate host-parasite interaction.

## Results and discussion

In pooled whole-genome sequencing of 600 infected and 600 uninfected 13–16-R1 snails (298x and 333x coverage, respectively), a single genomic region showed by far the greatest difference in allele frequencies between pools (*Figure 1A*; *Supplementary file 1A*). The highest outliers occurred in a 450 kb section of LG XII, here called PTC2. Genetic divergence between pools ($F_{ST}$) at numerous PTC2 variants exceeds 0.1, a value unobserved among one billion simulated neutral variants, which is therefore significant even if corrected for the nearly 7 million empirical variants examined (p<0.01). Many variants even show $F_{ST}$ over 0.2, more than twice the $F_{ST}$ at sod1 and RADres. By subsequently genotyping indel polymorphisms at PTC2 in individual snails, we observed three alleles at intermediate frequency (R: 44%, S1: 24%, and S2: 32%). Infection was rare for RR homozygotes (12.9%), and much more common for S1S1 (75.3%) and S2S2 (29.6%) homozygotes, a difference in infection odds of over 15-fold (i.e. infection odds of 0.15 vs. 3.0; *Figure 1B*). Heterozygotes showed intermediate phenotypes. There was weak partial dominance of S1 over R (observed Clopper-Pearson 95% confidence interval of infection probability for S1R = 51.0–61.8%; expected intermediate phenotype = 44.1%), such that relative to RR, carrying an S1 allele increases the odds of infection 5.9-fold (p=6 × $10^{-42}$) while a second S1 allele further increases the odds of infection 2.7-fold for a 15.9-fold difference (p=1 × $10^{-4}$). The S2 allele acts additively, such that each S2 allele increases the odds of infection 1.5-fold (p=6 × $10^{-5}$). We confirmed the PTC2 signal using an independent set of 392 snails from 13-16-R1 that had previously been phenotyped (*Tennessen et al., 2015a*) (p=7 × $10^{-12}$ for R vs. S1; p=4 × $10^{-5}$ for R vs. S2; *Figure 1—figure supplement 1*). These snails had also been genotyped at sod1 and RADres, revealing that all three loci had significant independent associations when included together in the same model (p≤$10^{-4}$ for each), with no evidence for epistasis (p>0.05 for interaction terms). Segregating variation at PTC2 has a stronger association with odds of infection than that of any other known *B. glabrata* locus (*Tennessen et al., 2015a*; *Tennessen et al., 2015b*). The BS-90 RAPD marker (*Knight et al., 1999*) is only 5 Mb and 23 cM from PTC2 (*Figure 1—figure supplement 2*; *Tennessen et al., 2017*). This marker is predicted to be 17 cM (range 6–33 cM) from a causal locus, which could therefore plausibly be PTC2 (*Supplementary file 1B*).

Using PacBio whole-genome assemblies from snails homozygous for each of the three PTC2 alleles (*Supplementary file 1C*), we find striking sequence and structural divergence among the haplotypes (*Figure 2*). Alignable regions show 3.3% nucleotide divergence on average (SD = 2.1%). A majority of PTC2 sequence shows no similarity among alleles; the percentage of sequence that could

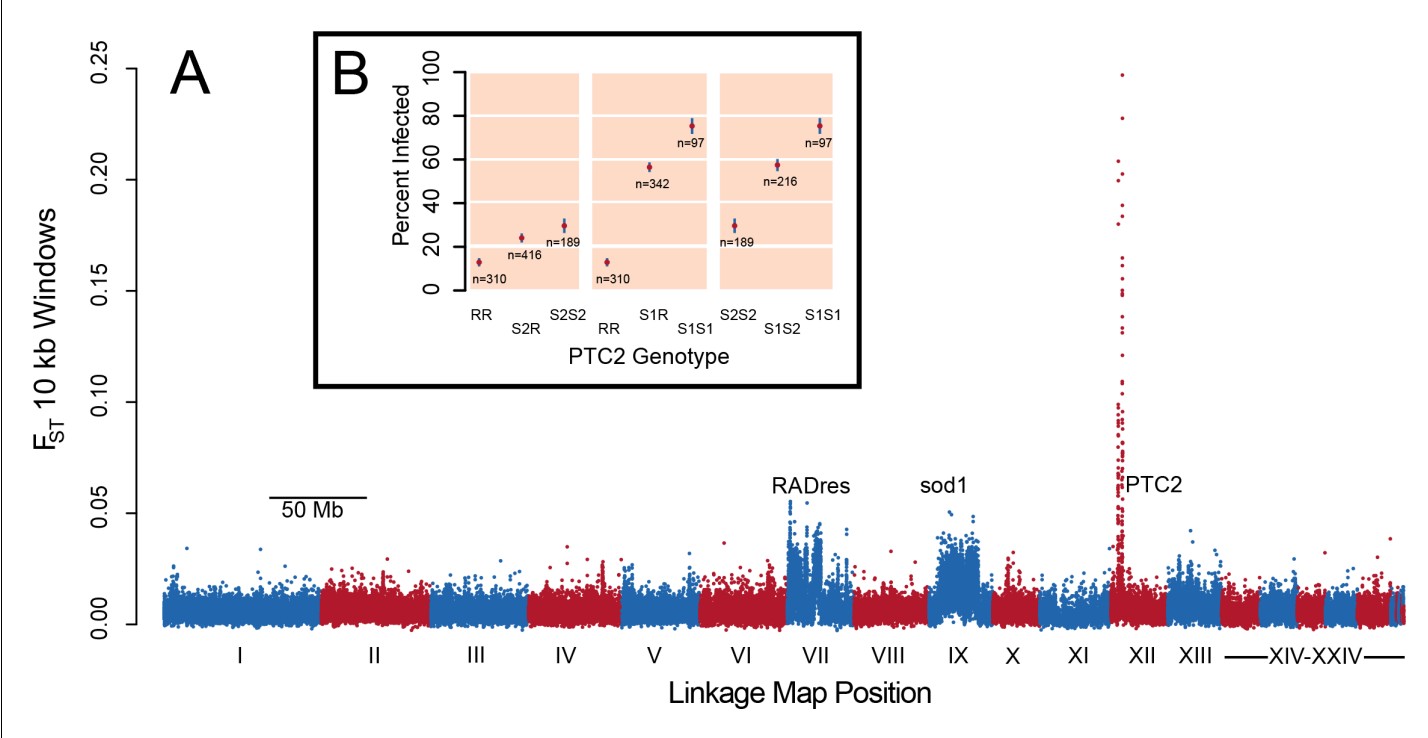

**Figure 1.** A region on Linkage Group XII displays a major association with infection risk (*Figure 1—source data 1*). (A) Genetic divergence ($F_{ST}$) between infected and uninfected snail pools in 10 kb windows, for variants on contigs arranged by linkage map position (*Tennessen et al., 2017*). The strongest signal is from PTC2 on Linkage Group XII, far exceeding the signal of known regions RADres and sod1 which reflect wide haplotype blocks (*Tennessen et al., 2015a*). (B) PTC2 genotypes are associated with infection outcome (*Source code 1*). Genotypes are displayed for all three pairs of alleles, revealing a substantial difference between R and S1 (center panel), and an intermediate signal of S2 relative to the others (left and right panels). The online version of this article includes the following source data and figure supplement(s) for figure 1:

**Source data 1.** Genetic divergence between infected and uninfected snail pools in 10 kb windows.
**Figure supplement 1.** Validation of PTC2 association with infection (*Source code 1*).
**Figure supplement 2.** Dot plot of tiled contigs from assemblies *homR* and *homS1* at the start of LG XII.

even be aligned ranged from 12.9% (R onto S2) to 40.0% (S1 onto R). All three PTC2 haplotypes harbor unique insertions tens of thousands of bp in size, some of which contain complete coding genes, such that each genotype carries a distinct combination of genes (*Figure 2*). Shared orthologous genes at PTC2 show many nonsynonymous differences and in some cases homology can only be identified at the protein, not DNA, level (*Figure 2—figure supplement 1*; *Figure 3*). This degree of polymorphism is unusually high for conspecific haplotypes in most genomic regions in any taxon (*Leffler et al., 2012*). In contrast to PTC2, 89.5% of sequence on other contigs can be aligned between assemblies, with a mean of 0.4% nucleotide divergence. It is not obvious how a chromosomal rearrangement (e.g. inversion) could maintain more than two distinct haplotypes, and in any case we see no evidence for one in our assemblies.

Using RNA-Seq data from homozygotes of each genotype to identify expressed genes, we fully annotated PTC2 (*Supplementary file 1D,E*). Of the eleven PTC2 genes, eight are predicted to be TM1 genes, including all five genes that are shared among the three haplotypes (genes 1, 2, 4, 5, and 9; *Figure 2*; *Figure 3*). Of the three non-TM1 genes, gene 3 and gene 11 show homology to TM1 genes 2 and 8, respectively, but without the TM1 domains. Gene 6 contains a conserved protein domain of unknown function (DUF2732). Only 11% of *B. glabrata* genes are TM1 genes, so they would be unlikely to constitute eight of eleven genes by chance alone ($p<10^{-5}$). PTC2 TM1 genes are all between 166 and 530 codons, have TM1 domains that are displaced from the N-terminus (*Figure 3—figure supplement 1*), and like the rest of PTC2 they are highly polymorphic (*Figure 3*), with amino acid level divergence exceeding 50% in several cases. Sequences similar to both R and S haplotypes are present in other *B. glabrata* populations without admixed histories (genomic and

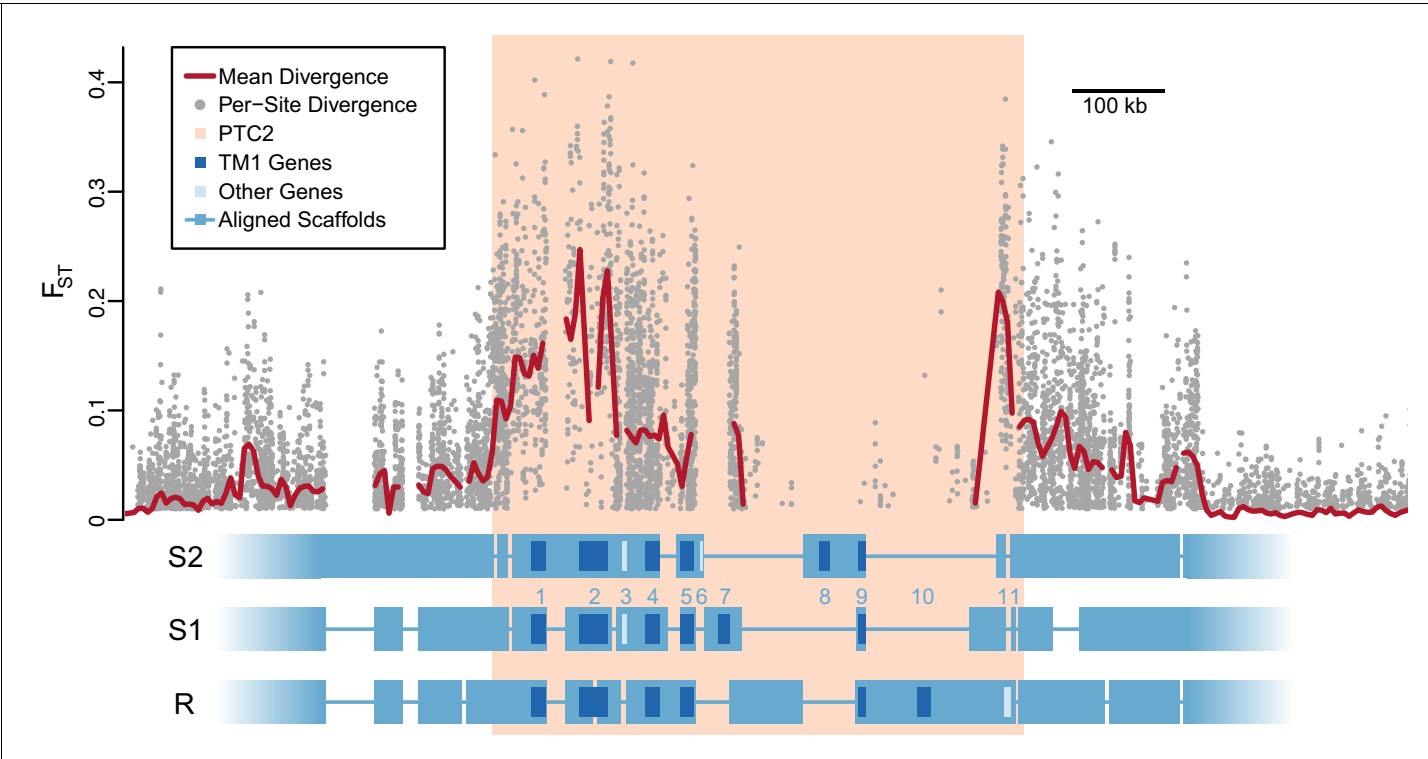

**Figure 2.** Divergent haplotypes of Polymorphic Transmembrane Complex 2 (PTC2). As in **Figure 1A**, genetic divergence ($F_{ST}$) between infected and uninfected snail pools is shown, here both for individual variants (grey circles, only $F_{ST} \geq 0.01$ shown) and as mean values for sliding windows of 10 kb (red line). $F_{ST}$ is undefined for regions present on only one haplotype. PTC2 (peach) is defined as the 450 kb region containing all windows for which mean $F_{ST}$ exceeds 0.1. Within PTC2, the three haplotypes (R, S1, and S2) are aligned with multi-kilobase indels and genes indicated. Assemblies are comprehensive and alignment gaps represent annotated indels, not missing data. PTC2 is characterized by extensive sequence divergence, including large indels containing entire genes (numbered), and is enriched for single-pass transmembrane (TM1) loci (dark blue).

The online version of this article includes the following figure supplement(s) for figure 2:

**Figure supplement 1.** Dot plots for haplotypes R and S1, R and S2, S1 and S2, and each haplotype against itself.

**Figure supplement 2.** Genetic divergence ($F_{ST}$) between infected and uninfected snail pools in 10 kb windows (red dots), in the vicinity of PTC2 (peach box), for reads aligned to all three genome assemblies (contigs = blue lines).

**Figure supplement 3.** Alignment of *homR* contigs R-35 and R-304.

**Figure supplement 4.** Alignment of *homS2* contigs and raw PacBio reads to reference genome *BglaB1* contig 13.

transcriptomic sequence from Brazilian strain BB02, **Adema et al., 2017**; transcriptomic sequence from Guadeloupe, **Tennessen et al., 2015b**; **Figure 3—figure supplement 2**), suggesting that they co-occur throughout the species range and the polymorphism is old. Synonymous divergence among alleles is higher than nonsynonymous, and a phylogeny of concatenated genes shows 24% synonymous divergence from the midpoint root. Thus, haplotypes are more consistent with an ancient origin (24 million years assuming a neutral mutation rate of $10^{-8}$ per year) rather than recent divergence via selection for protein diversity. Other than the transmembrane segment, these genes contain no known protein domains or homology to sequences outside of gastropods, nor are they homologous to GRC genes. Some show homology to each other and/or to other genes near PTC2 or elsewhere in the genome, but amino acid level sequence similarity among paralogs is low (<50%). The phenotypic effects of individual genes and polymorphisms will be an exciting subject for future work involving knockdowns or knockouts (**Allan et al., 2017**; **Abe and Kuroda, 2019**) and/or additional RNA-Seq from multiple individuals allowing quantification of expression differences.

Both GRC and PTC2 suggest a model of snail-schistosome interaction via molecular recognition (either of the parasite by the host, or of the host by the parasite) that is mediated by TM1 gene polymorphism. Across metazoans, TM1 genes often play a role in immunological recognition, and include B- and T-cell receptors, Toll-like receptors, major histocompatibility complex genes, and similar host defense genes (**Pahl et al., 2013**). Other polymorphic clusters of host transmembrane genes

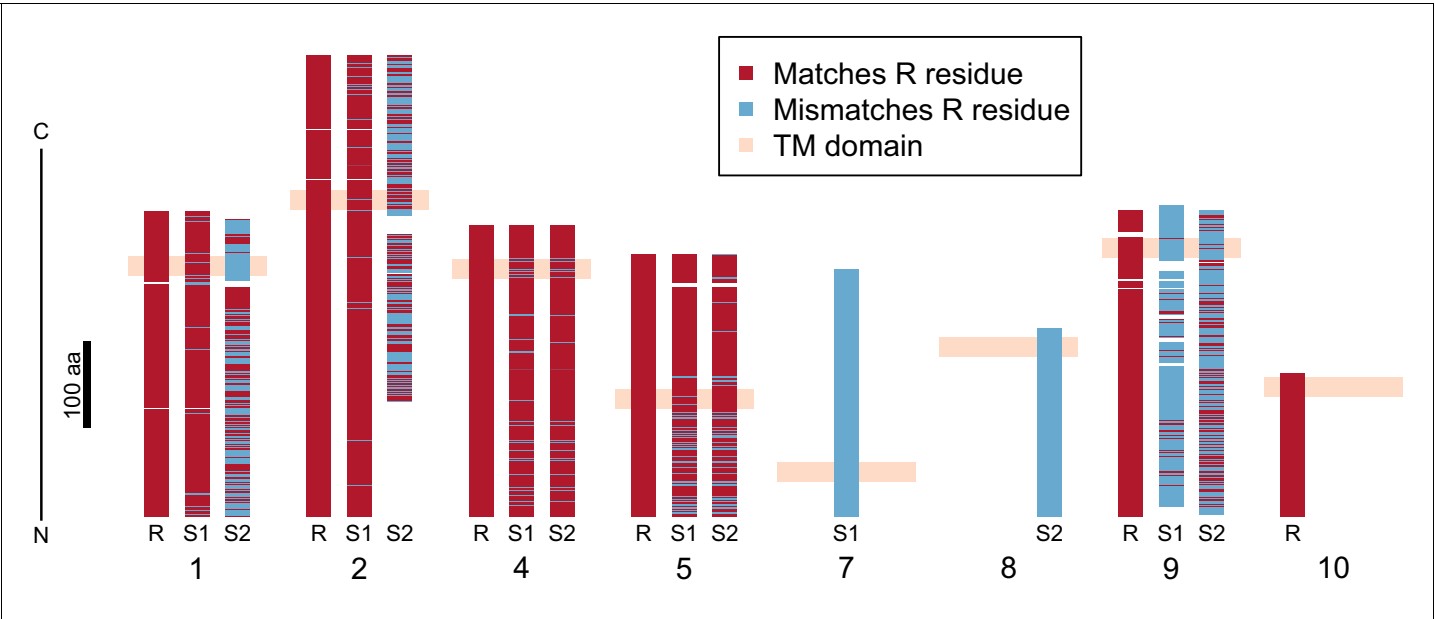

**Figure 3.** Divergence of PTC2 single-pass transmembrane (TM1) genes. For each of eight TM1 genes (numbered as in *Figure 2*; *Supplementary file 1D*), the protein product is shown, including the transmembrane domain (peach). Orthologous alleles are aligned. Amino acid residues that differ from the R allele are shown in blue. Uncorrected protein sequence divergence between orthologous alleles ranges from 3% to 85%. Three genes (7, 8, and 10) are present only on a single haplotype.

The online version of this article includes the following figure supplement(s) for figure 3:

**Figure supplement 1.** Distribution of the transmembrane domain within the protein sequence for all single-pass transmembrane (TM1) genes in *B. glabrata*, with positions of the PTC2 genes indicated (blue lines, averaged across alleles; numbered as in *Figure 3*).

**Figure supplement 2.** Unrooted nucleotide phylogenies of the coding sequence of the two most conserved PTC2 TM1 genes, including RNA sequence from Caribbean population GUA and genomic and RNA sequence from Brazilian strain BB02.

are used by parasites as receptors for host recognition and invasion (e.g. human glycophorins and *Plasmodium*; *Malaria Genomic Epidemiology Network et al., 2017*), and at least one of the GRC TM1 genes controls shedding of *S. mansoni* cercariae (*Allan et al., 2017*). One PTC2 TM1 gene is present only on the R haplotype and is an obvious candidate if it functions to recognize the parasite. However, allelic divergence among shared genes could also be important, and an R-specific gene alone would not explain the difference between S1 and S2. In contrast to GRC, in which a completely dominant allele confers resistance, all three alleles in PTC2 differ in their susceptibility, and allelic associations are additive or show partial dominant susceptibility (*Figure 1B*). This pattern suggests that multiple loci along the haplotypes may jointly contribute to phenotype by interacting with different combinations of parasite molecules such as SmPoMucs (*Roger et al., 2008*) or other glycoproteins (*Allan and Blouin, 2018*) to determine the outcome of infection.

As with GRC, we suspect non-neutral host-parasite coevolutionary processes have shaped sequence polymorphism at PTC2. The inferred ancient origin (>20 million years) of PTC2 is inconsistent with a neutral coalescent process. Because 13–16-R1 is admixed from geographically isolated populations, we can't infer natural allele frequencies or compare the site frequency spectrum to a neutral expectation for a randomly-mating population, as was possible for GRC (*Tennessen et al., 2015b*), though these allelic lineages do segregate in natural populations (*Figure 3—figure supplement 2*). Therefore, while the remarkable structural and nonsynonymous polymorphism appears adaptive, it is difficult to distinguish among plausible scenarios including overdominance (*Woolhouse et al., 2002*), negative-frequency dependent selection (*Woolhouse et al., 2002*; *Koskella and Lively, 2009*; *Bento et al., 2017*), adaptive introgression from distantly related species (*Hedrick, 2013*), or selection for an epistatically-interacting supergene (*Thompson and Jiggins, 2014*). Introgression appears unlikely, as it would have had to occur twice independently to generate three distinct haplotypes, and all of the closest relatives of *B. glabrata* occur allopatrically in Africa (*DeJong et al., 2001*). Therefore, the most plausible explanations involve some form of long-term

balancing selection. Although *S. mansoni* is not native to the neotropics (*Desprès et al., 1993*), selection may have been driven by other trematodes, a clade that has ubiquitously infected snails for millions of years (*Blair et al., 2001*) and which can be a strong selective force favoring rare alleles (*Koskella and Lively, 2009*). Schistosomes castrate snails (*Faro et al., 2013*) but wild snails show no sign of evolving universal resistance, suggesting that the R haplotype is either specific to parasite genotype or else costly to fitness in some undetected manner. The R allele has persisted within the 13–16-R1 population for decades in the absence of challenge by parasites, so any fitness cost must be relatively weak or context-dependent.

We have thus far only observed an effect on one parasite strain, PR-1, precluding inferences about gene-for-gene interaction. Nevertheless, a system of polymorphic matching alleles could explain the substantial schistosome-strain by snail-strain interaction in compatibility that is often observed (*Richards and Shade, 1987*; *Richards et al., 1992*; *Knight et al., 1999*; *Webster et al., 2004*; *Webster and Woolhouse, 1998*; *Theron et al., 2014*), including schistosome-infection dose-response curves that fit a simple phenotype-matching model (*Theron et al., 2008*; *Theron et al., 2014*). If more than one PTC2 gene contributes to resistance, then synergistic interactions among these genes and other unlinked loci could begin to explain the pronounced variation in host-parasite compatibility (*Schulenburg et al., 2007*). In other invertebrates, highly polymorphic haplotypes can be major-effect loci for infection and show striking coevolutionary signatures including variable presence/absence of genes. For example, the PR-locus mediating bacterial resistance in *Daphnia* via matching-allele interactions also features haplotypes of vastly different sizes (differences > 60 kb at both this locus and PTC2) with large non-homologous sections, and these contain glycosyltransferase genes that could mediate host-pathogen compatibility (*Bento et al., 2017*). Similarly, the APL1 immune factor impacting *Plasmodium* development in *Anopheles* consists of adjacent paralogs that differ in copy number among species and show extreme diversity within species (*Rottschaefer et al., 2011*; *Mitri et al., 2020*). Thus, while invertebrates lack the acquired immunity of vertebrates and its associated adaptive genetic variation (*Spurgin and Richardson, 2010*), their defenses can show similar nuance conveyed by molecular diversity (*Loker et al., 2004*; *Cerenius and Söderhäll, 2013*). However, the evolutionary consequences may not match those for vertebrate acquired immunity loci like the major histocompatibility complex or immunoglobulins, where sequence diversity per se tends to enhance immune effectiveness though perhaps at the cost of autoimmunity. For invertebrates, increased numbers of distinct immunogenetic sequences may not necessarily lead to increased resistance if parasites also use these sequences to recognize hosts or mount evasion strategies (*Schmid-Hempel, 2005*). If discarding immune genes is often as advantageous as gaining them, the result could be a patchwork of genes as observed at PTC2. A more appropriate vertebrate analog might be blood groups used by parasites to invade host cells, for which haplotypic differences often include loss of functional genes, and diversity has been maintained for millions of years by balancing selection (e.g. Dantu group, *Malaria Genomic Epidemiology Network et al., 2017*; ABO group, *Ségurel et al., 2013*). The generation and maintenance of such divergent haplotypes remain to be fully explained and could reflect long-term fluctuating selection among alleles with different combinations of specificity and cost (*Seger, 1988*; *Ashby and Boots, 2017*).

As with mosquitoes (*Marshall et al., 2019*), a promising strategy for disease control involves recruiting the natural immunogenetic variation of vectors (*Reardon, 2016*). The successful implementation of CRISPR/Cas9 in gastropods (*Abe and Kuroda, 2019*) will facilitate the creation of genetically modified snails having enhanced immunity to block disease transmission (*Maier et al., 2019*). As large-effect loci, the TM1 clusters are excellent candidates to target in such efforts. However, more work is needed to characterize the functional effects of these genes, as well as the molecular and evolutionary dynamics between hosts and parasites. For example, if host polymorphism is adaptive, it may not be readily replaced in natural populations. Furthermore, gene-by-gene interactions between snail and schistosome genotypes could permit the rapid evolution of parasite counterstrategies. In the context of ancient trench warfare coevolution, it is unlikely that a universally resistant snail could be generated by a single genetic change, although successive changes could enhance parasite resistance enough to impact patterns of transmission (*Theron et al., 2014*). As an alternative to genetic modification, future work could leverage the hypothesis that the snail TM1 proteins bind to key schistosome molecules that mediate invasion of the host. One could use TM1 proteins to find such molecules, as with snail fibrinogen related proteins and schistosome SmPoMucs

(*Roger et al., 2008*), or GRC and galactose (*Allan and Blouin, 2018*). More broadly, clusters of immune recognition loci with elevated functional diversity have long been used to track and predict patterns of adaptive variation across populations and species (*Sommer, 2005*; *Spurgin and Richardson, 2010*). Thus, we anticipate that this class of genes will play a central role in disease control as the molecular aspects of vector biology are fully brought to bear on schistosomiasis.

# Materials and methods

## Key resources table

| Reagent type (species) or resource | Designation | Source or reference | Identifiers | Additional information |
|---|---|---|---|---|
| Biological sample (*Biomphalaria glabrata*) | 13–16-R1 | PMID:5050093; PMID:7299581 | 5050093: NIH-MH-cc-13-16-1; 7299581:13–16-R1 | Oregon State University population established by C. Bayne |
| Strain, strain background (*Schistosoma mansoni*) | PR-1 | PMID:5050093; PMID:7299581 | 5050093: NIH-Sm-PR$_1$; 7299581:PR-1 | |
| Sequence-based reagent | PB35_1696 k_F | This paper | PCR primer | GGTTCTCGCTTTTTATTGGCTTTTG |
| Sequence-based reagent | PB35_1696 k_R | This paper | PCR primer | TTAGACGCACCCAAGGATCTC |
| Sequence-based reagent | VB13_859 k_Fb | This paper | PCR primer | ACAAATGGGGCAGTTACACTGTTTAC |
| Sequence-based reagent | VB13_859 k_Rb | This paper | PCR primer | AGCGAAATGTGAGATTGGTTATGTTG |
| Sequence-based reagent | VB13_868 k_Fb | This paper | PCR primer | TCTTTTCACTAAAGCCGCACAAGTT |
| Sequence-based reagent | VB13_868 k_Rb | This paper | PCR primer | CCTACGTTCTCAATATCAACGGGAA |
| Software, algorithm | SimulatePools.pl | https://github.com/jacobtennessen/GOPOPS/ | Perl script | power analysis for pooled sequencing |
| Software, algorithm | MakeFreqTableFromPooledPileup.pl | https://github.com/jacobtennessen/GOPOPS/ | Perl script | estimates allele frequencies |
| Software, algorithm | FstFromJoinedFreqTablesWindow.pl | https://github.com/jacobtennessen/GOPOPS/ | Perl script | calculates mean $F_{ST}$ per window |
| Software, algorithm | ChopFastaStaggered.pl | https://github.com/jacobtennessen/MiSCVARS/ | Perl script | subdivides sequence data |
| Software, algorithm | AssessBlatChopped.pl | https://github.com/jacobtennessen/MiSCVARS/ | Perl script | summarizes sequence matches |

## Animals and Ethics

We used the Oregon State University population of 13–16-R1 that has been maintained as a large population (hundreds) since the mid-1970s (*Bonner et al., 2012*). 13–16-R1 is descended from snails collected in Brazil and Puerto Rico (*Richards and Merritt, 1972*; *Sullivan and Richards, 1981*) but its exact history is not entirely clear. Our population has been maintained in the absence of parasite exposure, and therefore under relaxed selective pressure in regard to parasite resistance.

We used mice to maintain the schistosome parasites and to produce miracidia for challenge experiments. Infection is through contact with inoculated water and involves minimal discomfort. Infected rodents are euthanized with $CO_2$ prior to showing clinical signs of disease and are dissected to recover parasitic eggs. Animal numbers were held to the minimum required for the research. Institutional approval: Oregon State University Animal Care and Use Protocols 4749 and 5115.

## Genome-wide scan of 13–16-R1

We challenged snails of the 13–16-R1 population with PR-1 miracidia, following previous methods (*Bonner et al., 2012*). In brief, we arbitrarily chose 1700 outbred juvenile snails (4–6 mm diameter), challenged them each with five miracidia, and classified them as infected or uninfected. About 40%

of snails became infected. From these, we randomly selected 600 infected and 600 uninfected snails for sequencing. These sample sizes were chosen based on a simulation of variants with minor allele frequencies $\geq$ 0.2, with copies randomly assigned to 600 infected and 600 uninfected individuals at the expected sequencing coverage depth (script SimulatePools.pl at https://github.com/jacobten-nessen/GOPOPS/), which revealed that $F_{ST}$ between simulated sequencing pools was unlikely to exceed 0.05 ($p<10^{-5}$) and very unlikely to exceed 0.1 ($p<10^{-9}$) and therefore we had substantial power to detect larger $F_{ST}$ differences. We divided the empirical pools into two technical replicates, and four pools (each combination of infected/uninfected and technical replicate) were sequenced across six lanes of the Illumina HiSeq 3000 (paired-end reads of 151 bp) at the Center for Genome Research and Biocomputing (CGRB) at Oregon State University (Illumina data at NCBI SRA, BioProject Accession PRJNA638474).

Infected snails contain DNA from *S. mansoni*, which could potentially generate false sequence variants correlated with resistance. To prevent this, we converted reads to FASTA format, used BLASTN (version 2.6.0) to identify reads that matched the *S. mansoni* reference genome (v. 5.2, *Berriman et al., 2009*) with an E-value cutoff of 1e-040, and then filtered all such reads, as well as their mate pairs, from all downstream analysis. Filtered FASTQ files, having had adapters removed with Cutadapt (version 1.15, *Marcel, 2011*) and trimmed with Trimmomatic (v. 0.30, *Bolger et al., 2014*; options: LEADING:20 TRAILING:20 SLIDINGWINDOW:5:20 MINLEN:50), were aligned using BWA version 0.7.12 (command: bwa mem -P -M -t 4; *Li and Durbin, 2009*) initially to reference genome *BglaB1* and ultimately to our PacBio assemblies (*Figure 2—figure supplement 2*). All reads marked as secondary alignments were filtered out of the sam files. We used SamTools version 1.3 (*Li et al., 2009*) to convert these to sorted bam files (commands: samtools view -bT; samtools sort) and generate pileup files (command: samtools mpileup -t DP -A). From these files, we estimated allele frequencies at each variant within each pool, and calculated $F_{ST}$ in overlapping 10 kb windows across the genome, using the scripts MakeFreqTableFromPooledPileup.pl (options: -a 0.1 and -d 15) and FstFromJoinedFreqTablesWindow.pl (default options), available at https://github.com/jacobten-nessen/GOPOPS/. We only considered windows with at least 20 single-nucleotide polymorphisms, in order to exclude associations that are supported by few variants and which are therefore likely to be spurious.

To more precisely estimate genotype-phenotype associations at the LG XII candidate region, we genotyped candidate loci from the region in individual snails. We designed primers for genotyping using Primer-BLAST (https://www.ncbi.nlm.nih.gov/tools/primer-blast/) on the consensus of our assemblies and used them for PCR amplification (*Supplementary file 1F*). These surrounded indels, such that after initial confirmatory sequencing in test samples, samples could be genotyped with PCR and gel electrophoresis alone. We genotyped the candidate locus in 1570 of the original 1700 phenotyped snails, including 1165 of the 1200 samples used in the genome-wide association study. Furthermore, in order to independently validate the candidate region, we also genotyped it in 392 snails (also 13–16-R1) from a set of 439 that had been phenotyped several years previously (*Tennessen et al., 2015a*). We tested for effects between genotype and phenotype using logistic regression, following our standard approach (*Tennessen et al., 2015a*; *R Development Core Team, 2020*; *Source code 1*). Specifically, we first coded infection as binary (1 or 0) and each allele as either additive ('add': 0, 1, or 2 copies of the allele), dominant ('dom': 1 or 0 for presence/absence of allele), or recessive ('rec': 1 if homozygous, 0 otherwise). We first confirmed an independent effect of both S alleles relative to the R allele with model glm(infection~S1add+S2add, family = binomial), and then we found the best-fitting parameter combination (minimum Akaike information criterion) which was model glm(infection~S1dom+S1rec+S2add, family = binomial). The positive effect of both S1dom and S1rec on infection odds was interpreted as partial dominance (i.e. increased susceptibility if the allele is present plus additional increased susceptibility for homozygotes). We tested for epistasis by first adding terms for RADres and sod1 (known to act additively; *Tennessen et al., 2015a*) to the model and then testing if interaction terms among loci were significant.

## PacBio sequencing and assembly

We generated three PacBio assemblies from inbred snail lines homozygous for the three PTC2 alleles. (*Supplementary file 1C*; assemblies NCBI Genome, BioProject Accession PRJNA639204). The first assembly (*homR*) used snail line R68, which is derived from 13-16-R1 and is highly resistant to *S. mansoni* strain PR-1, as described previously (*Tennessen et al., 2015a*). We pooled and

sequenced these snails in 15 SMRT cells (78x coverage) on the Pacific Biosciences Sequel I at the CGRB. We assembled the resulting raw sequences using the HGAP4/FALCON assembler (*Chin et al., 2016*; options: Genome Length 1 Gb, Seed coverage 30, Min Map Concordance 70). Similarly, the other two assemblies (*homS1* and *homS2*) were generated using the same methodology from snail lines i90 (6 SMRT cells, 58x coverage) and i171 (5 SMRT cells, 46x coverage), respectively. By default, we treat *homR* as the reference genome unless stated otherwise. To assign PacBio contigs to the existing linkage map (*Tennessen et al., 2017*), we aligned 46,023 fragments of 100 bp each from *BglaB1* (the published genome) that had previously been screened for uniqueness and used for targeted capture (*Tennessen et al., 2017*) using BLASTN (version 2.6.0) with default parameters. PacBio contigs were then assigned to linkage groups if at least one unique fragment from a mapped *BglaB1* contig aligned to it, and if at least 75% of these matching unique fragments pertained to the same linkage group. We thus assigned 1489 *homR* contigs to linkage groups, representing 635 Mb; these assignments were supported by a median of seven mapped fragments per *homR* contig, with an average of 96% of fragments per *homR* contig mapping to the same linkage group.

In the vicinity of PTC2, we assessed sequence similarity with dot plots. Each assembly was broken into overlapping 600 bp segments (script ChopFastaStaggered.pl at https://github.com/jacobtennessen/MiSCVARS/), which were tested for sequence similarity in pairwise comparison using BLAT (*Kent, 2002*; options: stepSize = 1 -minScore = 300) followed by script AssessBlatChopped.pl (at https://github.com/jacobtennessen/MiSCVARS/). To estimate average genomic sequence similarity outside of PTC2, we used BLASTN (version 2.6.0) to identify pairs of orthologous contigs between *homR* and *homS1* (our two best assemblies, which should represent random samples of 13–16-R1 in regions unlinked to PTC2), and performed a similar BLAT comparison for all such pairs in which both contigs were over 2 Mb. For assemblies *homR* and *homS2*, PTC2 is split between two contigs each (*Supplementary file 1C*). We manually combined these contigs into continuous haplotypes. For *homR*, the ends of contigs R-35 and R-304 both align to each other with 99.9% similarity for 22 kb, indicating that they are in fact directly adjacent and the assembly algorithm was overly conservative in failing to join them (*Figure 2—figure supplement 3*). For the *homS2* contigs, raw reads aligning to contig S2-78 overlapped with raw reads aligning to contig S2-773, indicating a gap of only 12.6 kb which was confirmed by alignment to *BglaB1* (*Figure 2—figure supplement 4*).

## RNA-Seq and annotation

Although *BglaB1* is annotated, many genes were likely missed, especially those spanning multiple contigs. Furthermore, some PTC2 haplotypes may contain genes missing from the reference genome. Therefore, we performed RNA-Seq on snail lines homozygous for each of the three PTC2 haplotypes in order to identify all expressed proteins on each haplotype. Samples were prepared as described previously (*Tennessen et al., 2015b*). A single sample from each homozygous genotype was included in the same lane of the Illumina HiSeq 3000 at the CGRB (single-end reads of 151 bp; Illumina data at NCBI SRA, Bioproject Accession PRJNA639026). This single-sample approach precludes quantifying expression in a rigorous way (*Supplementary file 1E*), but not our goal of assembling transcriptomes for the purpose of annotation. We performed a de-novo annotation of each PTC2 haplotype. Each haplotype-specific RNA-Seq dataset was adapter and quality trimmed using Cutadapt (version 1.15, *Marcel, 2011*; options: -q 15,10) and de-novo assembled into a transcriptome assembly using Trinity (*Grabherr et al., 2011*; default assembly parameters). Transcriptome assemblies were reduced to longest open reading frames using TransDecoder (*Haas et al., 2013*) by first identifying the longest open reading frames (TransDecoder.LongOrfs), then using BLAST (*Altschul et al., 1990*) to map the longest open reading frames to the UNIPROT (*UniProt Consortium, 2019*) gastropod protein database (options: -max_target_seqs 1 -outfmt 6 -evalue 1e-5), and finally by predicting protein sequences from the assembled transcripts (TransDecoder.Predict). AUGUSTUS (*Stanke et al., 2004*) gene prediction training model was built from the UNIPROT *Biomphalaria* dataset. BUSCO (*Seppey et al., 2019*) was run on the *homS1* genome assembly for use across all assemblies. Single copy orthologs found by BUSCO were used to make the SNAP (*Korf, 2004*) gene prediction training set. A snail-specific repeat library was constructed using data from *BglaB1* (*Giraldo-Calderón et al., 2015*; *Adema et al., 2017*; https://www.vectorbase.org), and mollusca-specific repeats from Repbase (*Bao et al., 2015*), and these repeats were then masked using RepeatMasker (*Smit et al., 2013*). De-novo gene prediction was run with MAKER

(*Cantarel et al., 2008*) on the repeat-masked genome assembly using the TransDecoder reduced transcriptome assembly as EST evidence, the UNIPROT *Biomphalaria* proteins as protein evidence, and de-novo gene prediction was conducted using SNAP and AUGUSTUS using the constructed prediction models. We used these automated annotations, along with predictions from genomic sequence from GENSCAN (*Burge and Karlin, 1997*), and putative orthologous transcripts in the reference genome project (*Giraldo-Calderón et al., 2015*; *Adema et al., 2017*) identified with BLASTN (version 2.6.0), to guide manual alignment of RNA-Seq reads. Putative coding genes were rejected and subsequently ignored if they showed homology to transposable elements (e.g. RNA transcriptase or transposase) which are very abundant in the snail genome, if the open reading frame was less than 100 codons, or if the sequence could not be confirmed via manual alignment of RNA-Seq reads (sequences in NCBI GenBank, Accessions MT787302-MT787323). Secondary structure was predicted using TMHMM v. 2.0 (*Sonnhammer et al., 1998*).

To investigate the phylogenetic history of alleles, we first focused on the coding sequence of the two most conserved TM1 genes (4 and 5) as these could be aligned the most unambiguously. We searched for similar sequences in the genomic and transcriptomic data of reference genome *BglaB1* generated from BB02 (*Giraldo-Calderón et al., 2015*; *Adema et al., 2017*; https://www.vectorbase.org) and RNA-Seq data from Guadeloupe population GUA (*Tennessen et al., 2015b*; Bioproject Accession PRJNA264063). We conducted phylogenetic analysis using RAxML (options: -N 100 m GTRCAT; *Stamatakis, 2006*) and displayed trees with FigTree version 1.4.4 (http://tree.bio.ed.ac.uk/software/figtree/). We also aligned and concatenated coding sequence from the 13–16-R1 alleles of the five genes present on all three haplotypes and used SNAP version 2.1.1 (https://www.hiv.lanl.gov/content/sequence/SNAP/SNAP.html; *Korber, 2000*) to calculate nonsynonymous and synonymous divergence among alleles, and to infer synonymous site divergence from the midpoint root of a three-taxon neighbor-joining tree.

## Acknowledgements

We thank Leeah Whittier, Ryan Wilson, and Haley Hutcheson for snail husbandry assistance, and Angela Early for helpful comments. This work was partially supported by National Institutes of Health (http://www.nih.gov/) grant AI143991 to M Blouin.

## Additional information

### Funding

| Funder | Grant reference number | Author |
|---|---|---|
| National Institutes of Health | AI143991 | Michael Scott Blouin |

The funders had no role in study design, data collection and interpretation, or the decision to submit the work for publication.

### Author contributions

Jacob A Tennessen, Data curation, Formal analysis, Visualization, Writing - original draft, Writing - review and editing; Stephanie R Bollmann, Data curation, Formal analysis, Investigation, Project administration, Writing - review and editing; Ekaterina Peremyslova, Clint Sergi, Investigation; Brent A Kronmiller, Bulut Hamali, Formal analysis; Michael Scott Blouin, Conceptualization, Resources, Supervision, Funding acquisition, Project administration, Writing - review and editing

### Author ORCIDs

Jacob A Tennessen  https://orcid.org/0000-0002-5015-4740
Michael Scott Blouin  http://orcid.org/0000-0002-8439-2878

### Ethics

Animal experimentation: We used mice to maintain the schistosome parasites and to produce miracidia for challenge experiments. Infection is through contact with inoculated water and involves minimal discomfort. Infected rodents are euthanized with $CO_2$ prior to showing clinical signs of disease

and are dissected to recover parasitic eggs. Animal numbers were held to the minimum required for the research. Institutional approval: Oregon State University Animal Care and Use Protocols 4749 and 5115.

### Decision letter and Author response
Decision letter https://doi.org/10.7554/eLife.59395.sa1
Author response https://doi.org/10.7554/eLife.59395.sa2

## Additional files

### Supplementary files
• Source code 1. R script for analyzing individual snail genotypes and phenotypes with logistic regression.

• Supplementary file 1. Supplementary tables.

• Transparent reporting form

### Data availability
All data not included in the manuscript are available at NCBI. PacBio genome assemblies are available under BioProject Accession PRJNA639204. Pooled whole-genome sequencing reads are available under BioProject Accession PRJNA638474. RNA-Seq reads are available under BioProject Accession PRJNA639026. Assembled transcripts are on Genbank, Accessions MT787302-MT787323.

The following datasets were generated:

| Author(s) | Year | Dataset title | Dataset URL | Database and Identifier |
|---|---|---|---|---|
| Tennessen JA, Bollmann SR, Peremyslova E, Kronmiller BA, Sergi C, Hamali B, Blouin MS | 2020 | PacBio assemblies of Biomphalaria glabrata snails derived from lab strain 13-16-R1 | http://www.ncbi.nlm.nih.gov/bioproject/?term=PRJNA639204 | NCBI BioProject, PRJNA639204 |
| Tennessen JA, Bollmann SR, Peremyslova E, Kronmiller BA, Sergi C, Hamali B, Blouin MS | 2020 | Genome-wide association study of Biomphalaria glabrata resistance to Schistosoma mansoni | http://www.ncbi.nlm.nih.gov/bioproject/?term=PRJNA638474 | NCBI BioProject, PRJNA638474 |
| Tennessen JA, Bollmann SR, Peremyslova E, Kronmiller BA, Sergi C, Hamali B, Blouin MS | 2020 | RNAseq of Biomphalaria glabrata 1316 inbred lines | http://www.ncbi.nlm.nih.gov/bioproject/?term=PRJNA639026 | NCBI BioProject, PRJNA639026 |

The following previously published dataset was used:

| Author(s) | Year | Dataset title | Dataset URL | Database and Identifier |
|---|---|---|---|---|
| Adema CM, Warren W, Wilson RK, Hillier LW, Minx P | 2017 | Biomphalaria glabrata genome BglaB1 | https://www.ncbi.nlm.nih.gov/genome/?term=APKA00000000.1 | NCBI Genome, APKA00000000.1 |

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
