## [Decision Letter]

**Acceptance summary:**

Your study investigates the genetic bases of snail resistance to schistosomes. Schistosomiasis is considered by the WHO as one of the most (after malaria) socioeconomically devastating parasitic disease. Your study showed extreme divergence between three haplotypes, that explain variation in resistance, a major breakthrough in a system hardly accessible to many of the genetic tools commonly used in other systems. The work constitutes a basis for more detailed studies in natural populations and other host-parasite combinations. In the long run it may contribute to reducing the burden of schistosomiasis.

**Decision letter after peer review:**

Thank you for submitting your article "Clusters of polymorphic transmembrane genes control resistance to schistosomes in snail vectors" for consideration by *eLife*. Your article has been reviewed by three peer reviewers, one of whom is a member of our Board of Reviewing Editors, and the evaluation has been overseen by Dominique Soldati-Favre as the Senior Editor. The following individual involved in review of your submission has agreed to reveal their identity: Yann Bourgeois (Reviewer #3).

The reviewers have discussed the reviews with one another and the Reviewing Editor has drafted this decision to help you prepare a revised submission.

Tennessen et al. investigated the genetic bases of snail resistance to schistosomes. Snails are the intermediate host (or vector) for schistosomes and therefore of high medical importance. Schistosomiasis is considered by the WHO as one of the most (after malaria) socioeconomically devastating parasitic disease. The current study examines genome-wide polymorphism and differentiation between snails susceptible and resistant to infection by schistosomes. Using pool-sequencing, they identify a region of high differentiation between resistant and susceptible genotypes. They genotype most of the same individuals that were included in the pools and reveal a system with one locus and three haplotypes (R, S1, S2) showing varying in the degree of protection they confer against infection. Using PacBio sequencing, they also show extreme divergence between the three haplotypes, to the extent that alignment is impossible in some regions. RNA-sequencing is used to characterize the region further. The evolutionary mechanisms maintaining this polymorphism remains unknown. The work constitutes a basis for more detailed studies in natural populations and other host-parasite combinations. As a long term perspective, the authors suggest that the knowledge about this genomic region can contribute to disease control.

The manuscript has been reviewed by three specialists. All three agree that this is a major step forward. However, all of them raise important points that need to be addressed in a revised version (with a point by point rebuttal letter), before a final decision can be reached. The main points are summarized here, the detailed reviews are below.

As it stands, the paper is of little interest for people outside the "schistosome world". While schistosomes are certainly important in their own right, a bit more effort could be invested into making this study more accessible to a wider audience.

Many aspects of the analysis, including the statistics, are not clear. More details and care in presentation are needed. Scripts should be fully available.

Even so, one reviewer stated that "The discussion remains cautious", there are numerous places where this is not the case (see reviewer #2). It is important present the results and their interpretation in an adequate way without overstating.

The paper hints at the idea that the mechanisms maintaining this extreme polymorphisms is host-parasite coevolution with balancing selection. Without much more work, there are a few further analysis that might be undertaken to support this better. This would also allow a link to other literature reporting on the role of balancing selection in the framework of host-parasite interactions.

Reviewer #1:

This study describes the finding of a genomic region for resistance in the schistosome vector *Biomphalaria*. The study includes a series of experiments including phenotypic screens, poolseq, re-sequencing and RNA-Seq. Technically the study is mostly fine. I have a few points I would like to see in a revised version.

1) The bioinformatic analysis of the data is described in a way that it would not be possible to trace the results. Parameter setting, threshold value etc. are not reported. Just naming the software used in not sufficient. Also, many intermediate steps are not reported.

2) The authors mention another study that reported a similar complex genomic region providing resistance to a parasite. They hint that invertebrates may differ in the way diversity is created with regard to resistance to parasites. Are there more such cases? It would be nice to compare the new results directly with the earlier study and discuss the similarities and differences.

3) The entire article is very Schistsome/*Biomphalaria* centred and unusually rich in self-citations. For people not working in this field, the article is not appealing. I would be happy to see much more efforts of the authors to widen their scope and make the study of interest for other people studying host-parasite interactions, at least those being interested in invertebrate hosts.

4) The dot plots in Figure 2B are impressive. It would be good to add dot plots of S1 against S1, R against R and S2 against S2. This would be helpful for a better understanding of the structure of the genomic region.

Reviewer #2:

The authors present an association study for resistance to a Schistosome parasite in *Biomphalaria* snails. The short report consists of the association study, a careful genome assembly of the resistant and two susceptible strains and annotation of those strains to compare genes present in the region. The manuscript stops short of any attempt to validate that the PTM2 region plays a major role in infection outcomes.

1) As written, the manuscript assumes a significant familiarity with the system that is unlikely for a broad audience. As a methods last style manuscript, much was left unclear until the end of the paper: how were individuals infected?, what is the nature of the population used? etc. More background on the system (previous founding about PTM1, etc. would be useful. I realize that any such expansion might jeopardize the "short report" status, but the increased clarity might outweigh the benefit of brevity.

2) While I understand that these snails are not a genetic model system with abundant tools to do validations of candidate loci, I feel that the language used is often too definitive, suggesting that the association is causal. For example, in the figure legend 1 "(B) PTC2 genotypes determine infection outcome." This is false. PTC2 genotypes are associated with infection outcome. Care should be taken to not oversell the association.

3) The annotations presented in Figure 3 are both interesting and a mess. Actually, they are partly interesting because they are a mess. The genomic region is highly divergent among haplotypes. Given the apparent homogenization of the admixed backgrounds across the genome in the population, it is amazing how divergent this region continues to be. Both LG7 and LG9 appear to be entire chromosomes associated with the phenotype, but PTC2 is a relatively small region. Is introgression from another species a possibility? Also, the authors discuss the structure of the genes in those different regions, but is it possible to make any additional predictions about the associations from the gene content?

4) There is no discussion of how statistical significance is assessed in the association study. In the Materials and methods, simulations suggest that Fst as always less than 0.05, but that isn't presented in the main text. I don't suspect that any analysis would diminish peak at PTC2, but some indication of "how" significant that peak is would be useful. There are several methods to do this, but the permutations should be appropriate and have already been completed.

Reviewer #3:

Summary

In this work, Tennessen et al. conduct a comprehensive study of the genetic bases of resistance to schistosomes in snail vectors. They examine genome-wide polymorphism and differentiation between snails susceptible and resistant to infection by schistosomes in a population followed for more than 40 years. Using pool-sequencing, they first identify a region of high differentiation between the two categories. They genotype most of the same individuals that were included in the pools and reveal a system with one locus and three alleles (R, S1, S2) showing partial overdominance and varying in the degree of protection they confer against infection. Using PacBio sequencing, they also show extreme divergence between the three haplotypes, to the extent that alignment is impossible in some regions. Using RNA-sequencing of one individual per homozygous genotype, they reannotate each of the three haplotypes and reveal a large excess of TM1 genes in this region, displaying elevated amino-acid divergence and diversity. The exact evolutionary mechanisms maintaining such polymorphisms are difficult to pinpoint, given that only one parasite strain was used in infection experiments, but the work constitutes a solid basis for more detailed studies in natural populations and other host-parasite combinations.

Assessment

This is a compelling and impressive work. The signal is strong, and is unlikely to be an artefact. The work goes well beyond simply studying association between genetic and phenotypic variation, and uses a broad set of techniques (infection experiments, Pool-seq, individual genotyping, RNA-seq, PacBio assembly) to characterize the molecular bases of resistance. The discussion remains cautious, and takes care of not over-interpreting results.

The manuscript is already good in its current state, but I had a few comments detailed below:

General suggestions

As the authors state in the Discussion, it is difficult to determine what drives the maintenance of this polymorphism (which vector(s) is the actual selective pressure)? Which selective dynamic underlies this diversity? Is there any chance that mutational load be exposed at the homozygous state? However, it may be possible to obtain more insights about the evolutionary history of this region based on the data already available. For example, what is the diversity within the three divergent haplotypes? A way to assess this in more details could be using ancestral recombination graphs for a few selected individuals (for example by calling SNPs for the three reference individuals and running ARGWeaver (https://github.com/mdrasmus/argweaver) or Relate (https://myersgroup.github.io/relate/index.html)). This could give an idea of the age of alleles and genealogies in the relatively conserved regions that flank the highly divergent loci. This may also provide more evidence for long-term balancing selection.

The authors mention that they performed RNA-sequencing on a single individual for each of the three homozygous genotypes. Do the authors have any plan to examine in more details how gene expression changes depending on allele combinations? It may be interesting to obtain an even more detailed picture of regulatory and non-synonymous variation, and how dominance/recessivity impact it. I do not think that more sequencing is necessary for this paper, but maybe discuss a bit more the potential of such an experiment?

---

## [Author Response]

The manuscript has been reviewed by three specialists. All three agree that this is a major step forward. However, all of them raise important points that need to be addressed in a revised version (with a point by point rebuttal letter), before a final decision can be reached. The main points are summarized here, the detailed reviews are below.As it stands, the paper is of little interest for people outside the "schistosome world". While schistosomes are certainly important in their own right, a bit more effort could be invested into making this study more accessible to a wider audience.

We have added substantial text and new citations about invertebrate immunity and host-parasite coevolution more broadly. Details below.

Many aspects of the analysis, including the statistics, are not clear. More details and care in presentation are needed. Scripts should be fully available.

We now include links to multiple scripts available on Github and more details about methodological parameters and statistics. Details below.

Even so, one reviewer stated that "The discussion remains cautious", there are numerous places where this is not the case (see reviewer #2). It is important present the results and their interpretation in an adequate way without overstating.

We have revised the wording throughout to clarify that we have only found a genotype-phenotype association and we have not demonstrated a specific causal gene or polymorphism. Similarly, while we speculate about immunogenetic function and host-parasite coevolution, we emphasize that these inferences are not definitive. Details below.

The paper hints at the idea that the mechanisms maintaining this extreme polymorphisms is host-parasite coevolution with balancing selection. Without much more work, there are a few further analysis that might be undertaken to support this better. This would also allow a link to other literature reporting on the role of balancing selection in the framework of host-parasite interactions.

As we note below in response to reviewer 3, inferences about adaptive evolution are limited by the artificial history of this laboratory population and by the available data. Nevertheless, we have added some additional evolution analyses, including reporting on nonsynonymous and synonymous diversity, sequence similarity to distant paralogs, similarity to alleles in other populations (new figure supplement), and estimated age of the haplotypes based on synonymous site divergence. We have also added additional references on balancing selection mediated by host-parasite interaction. Details below.

Reviewer #1:This study describes the finding of a genomic region for resistance in the schistosome vector Biomphalaria. The study includes a series of experiments including phenotypic screens, poolseq, re-sequencing and RNA-Seq. Technically the study is mostly fine. I have a few points I would like to see in a revised version.1) The bioinformatic analysis of the data is described in a way that it would not be possible to trace the results. Parameter setting, threshold value etc. are not reported. Just naming the software used in not sufficient. Also, many intermediate steps are not reported.

We have added many more details to our analytical methods, including specific parameters, links to several scripts available on Github, and a new source code file supporting our logistic regression analysis. e.g. subsections “Genome-wide scan of 13-16-R1”, “RNA-Seq and Annotation” and the Key Resources Table.

2) The authors mention another study that reported a similar complex genomic region providing resistance to a parasite. They hint that invertebrates may differ in the way diversity is created with regard to resistance to parasites. Are there more such cases? It would be nice to compare the new results directly with the earlier study and discuss the similarities and differences.

We have added more background and citations about invertebrate immunity, including a new paragraph in the Introduction and we have expanded our comparison with other loci in other species that show similar patterns.

3) The entire article is very Schistsome/Biomphalaria centred and unusually rich in self-citations. For people not working in this field, the article is not appealing. I would be happy to see much more efforts of the authors to widen their scope and make the study of interest for other people studying host-parasite interactions, at least those being interested in invertebrate hosts.

As mentioned above, we have added more text about invertebrate hosts more generally. We are actually one of the only groups so far to publish genome wide association analyses on *Biomphalaria glabrata* resistance to schistosomes, and in previous work we found TM1 genes similar to those described here. So, many relevant citations come from our group. However, we have added numerous citations from other systems to enhance the general appeal of this manuscript.

4) The dot plots in Figure 2B are impressive. It would be good to add dot plots of S1 against S1, R against R and S2 against S2. This would be helpful for a better understanding of the structure of the genomic region.

We have added dot plots of each haplotype against itself. In response to reviewer 2, we have also converted this figure to a figure supplement.

Reviewer #2:The authors present an association study for resistance to a Schistosome parasite in Biomphalaria snails. The short report consists of the association study, a careful genome assembly of the resistant and two susceptible strains and annotation of those strains to compare genes present in the region. The manuscript stops short of any attempt to validate that the PTM2 region plays a major role in infection outcomes.1) As written, the manuscript assumes a significant familiarity with the system that is unlikely for a broad audience. As a methods last style manuscript, much was left unclear until the end of the paper: how were individuals infected?, what is the nature of the population used? etc. More background on the system (previous founding about PTM1, etc. would be useful. I realize that any such expansion might jeopardize the "short report" status, but the increased clarity might outweigh the benefit of brevity.

We have added more details about the population and the infection method to the Introduction. Background about the other loci including GRC (aka PTC1) is also included in the Introduction.

2) While I understand that these snails are not a genetic model system with abundant tools to do validations of candidate loci, I feel that the language used is often too definitive, suggesting that the association is causal. For example, in the figure legend 1 "(B) PTC2 genotypes determine infection outcome." This is false. PTC2 genotypes are associated with infection outcome. Care should be taken to not oversell the association.

We have reworded the language here and elsewhere to avoid implying causality of any particular gene or marker.

3) The annotations presented in Figure 3 are both interesting and a mess. Actually, they are partly interesting because they are a mess. The genomic region is highly divergent among haplotypes. Given the apparent homogenization of the admixed backgrounds across the genome in the population, it is amazing how divergent this region continues to be. Both LG7 and LG9 appear to be entire chromosomes associated with the phenotype, but PTC2 is a relatively small region. Is introgression from another species a possibility? Also, the authors discuss the structure of the genes in those different regions, but is it possible to make any additional predictions about the associations from the gene content?

We have added a new figure supplement (Figure 3—figure supplement 2) illustrating how this sequence divergence occurs throughout the species range. Thus, the PTC2 polymorphism is not an artifact of the admixed origin of 13-16-R1 and is likely to show more allelic divergence than most other genomic regions even in natural populations. We have previously noted that the signals on LG7 and LG9 appear to reflect large haplotype blocks (Tennessen et al., 2015a), possibly maintained by inversions. We now state this in the legend of Figure 1. The regions on LG7 and LG9 have been discussed previously (Tennessen et al., 2015a); it will be interesting to revisit those regions and identify candidate genes using the PacBio assemblies presented here, but such an analysis is beyond the scope of this manuscript. The PTC2 signal appears small on a genome-wide scale (Figure 1) but this still represents hundreds of kilobases. We now discuss why introgression is an unlikely explanation for the polymorphism (Results and Discussion paragraph four). We have added some additional analyses of the PTC2 genes (e.g. phylogenetic, Results and Discussion paragraph two; assessment of transmembrane domain position using simulations, legend Figure 3—figure supplement 1).

4) There is no discussion of how statistical significance is assessed in the association study. In the Materials and methods, simulations suggest that Fst as always less than 0.05, but that isn't presented in the main text. I don't suspect that any analysis would diminish peak at PTC2, but some indication of "how" significant that peak is would be useful. There are several methods to do this, but the permutations should be appropriate and have already been completed.

Our power analysis was previously only mentioned in the Materials and methods, but now we discuss it in more detail in the Results and Discussion. Specifically, using simulations, we show that an Fst value greater than 0.1 (our threshold for defining PTC2) is very unlikely to occur by chance in pooled sequencing reads (uncorrected p < 10^-9^; corrected p < 0.01).

Reviewer #3:[…]General suggestionsAs the authors state in the Discussion, it is difficult to determine what drives the maintenance of this polymorphism (which vector(s) is the actual selective pressure)? Which selective dynamic underlies this diversity? Is there any chance that mutational load be exposed at the homozygous state? However, it may be possible to obtain more insights about the evolutionary history of this region based on the data already available. For example, what is the diversity within the three divergent haplotypes? A way to assess this in more details could be using ancestral recombination graphs for a few selected individuals (for example by calling SNPs for the three reference individuals and running ARGWeaver (https://github.com/mdrasmus/argweaver) or Relate (https://myersgroup.github.io/relate/index.html)). This could give an idea of the age of alleles and genealogies in the relatively conserved regions that flank the highly divergent loci. This may also provide more evidence for long-term balancing selection.

We appreciate and share the reviewer’s enthusiasm for tracking down the evolutionary basis for the polymorphism. Unfortunately, any additional insight that can be gained from the available data is limited, for two reasons. First, our snails have a contrived demographic history that violates the assumptions of most tests for selection. Specifically, 13-16-R1 is descended from several distinct populations and its founding likely involved an extreme bottleneck, and our reference assemblies are derived from inbred lines generated from 13-16-R1. Second, three haploid sequences is too small of a sample size for an informative ancestral recombination graph. Thus, there is no biologically meaningful way to use ARGWeaver or Relate with this dataset. However, a few additional evolutionary analyses are possible, and we have added them to highlight the evidence for balancing selection. We have added a new brief analysis showing the average synonymous divergence since the common ancestor is 24%, indicating an ancient origin millions of years ago. Ancient haplotypes are a commonly noted signature of balancing selection, though the specific form of balancing selection is not known conclusively. This pattern is also consistent with introgression, though we note that this is unlikely (Results and Discussion paragraph four). We have also added a figure showing the polymorphism occurs naturally in populations from across the species range. We have no evidence for mutational load in homozygotes (e.g. no heterozygotes excess in violation of Hardy-Weinberg equilibrium).

The authors mention that they performed RNA-sequencing on a single individual for each of the three homozygous genotypes. Do the authors have any plan to examine in more details how gene expression changes depending on allele combinations? It may be interesting to obtain an even more detailed picture of regulatory and non-synonymous variation, and how dominance/recessivity impact it. I do not think that more sequencing is necessary for this paper, but maybe discuss a bit more the potential of such an experiment?

We agree with the reviewer that additional gene expression studies would be interesting but are outside the scope of the present manuscript. We now discuss the potential for future work in paragraph two of the Results and Discussion.